# Routine antibiotics for infants less than 6 months of age with growth failure/faltering: a systematic review

Aamer Imdad ,[1] Fanny F Chen,[2] Melissa François,[2] Momal Sana,[3] Emily Tanner-Smith ,[4] Abigail Smith,[5] Olivia Tsistinas,[5] Jai K Das ,[6] Zulfiqar Ahmed Bhutta[6,7]

[1]Division of Gastroenterology, Department of Pediatrics, SUNY Upstate Medical University, Syracuse, New York, USA
[2]Norton College of Medicine, SUNY Upstate Medical University, Syracuse, New York, USA
[3]Monash Health, Clayton, Victoria, Australia
[4]College of Education, University of Oregon, Eugene, Oregon, USA
[5]Health Science Library, SUNY Upstate Medical University, Syracuse, New York, USA
[6]Division of Women and Child Health, Aga Khan University, Karachi, Pakistan
[7]Global Child Health, Hospital for Sick Children Research Institute, Toronto, Ontario, Canada

**Correspondence to**
Dr Aamer Imdad;
imdada@upstate.edu

## ABSTRACT

**Objective** This systematic review commissioned by WHO aimed to synthesise evidence from current literature on the effects of systematically given, routine use of antibiotics for infants under 6 months of age with growth failure/faltering.

**Settings** Low-income and middle-income countries.

**Participants** The study population was infants less than 6 months of age with growth failure/faltering.

**Intervention** The intervention group was infants who received no antibiotics or antibiotics other than those recommended in 2013 guidelines by WHO to treat childhood severe acute malnutrition. The comparison group was infants who received antibiotics according to the aforementioned guidelines.

**Primary and secondary outcomes** The primary outcome was all-cause mortality, and secondary outcomes: clinical deterioration, antimicrobial resistance, recovery from comorbidity, adverse events, markers of intestinal inflammation, markers of systemic inflammation, hospital-acquired infections and non-response. The Grading of Recommendations Assessment, Development and Evaluation approach was considered to report the overall evidence quality for an outcome.

**Results** We screened 5137 titles and abstracts and reviewed the full text of 157 studies. None of the studies from the literature search qualified to answer the question for this systematic review.

**Conclusions** There is a paucity of evidence on the routine use of antibiotics for the treatment of malnutrition in infants less than 6 months of age. Future studies with adequate sample sizes are needed to assess the potential risks and benefits of antibiotics in malnourished infants under 6 months of age.

**PROSPERO registration number** CRD42021277073.

## STRENGTHS AND LIMITATIONS OF THIS STUDY

⇒ This WHO funded systematic review was conducted by following the standard methods of Cochrane Collaboration.

⇒ Even though data were available for the use of antibiotics in severely malnourished children 6–59 months of age, no randomised trials were found in infants less than 6 months of age.

⇒ The work was limited due to a lack of studies in this age group. Future studies with large sample size are needed to assess the efficacy and safety of antibiotics in malnourished infants less than 6 months of age.

## INTRODUCTION

The WHO and the UNICEF estimate that nearly 14 million children suffer from severe wasting (low weight for height) worldwide.[1] Infants less than 6 months of age are particularly vulnerable to the effects of inadequate nutrition. Higher mortality rates secondary to growth failure are seen in this age group compared with older infants and children.[2 3] Despite excess mortality risk and increasing prevalence of wasting in this population, limited studies exist to guide the management of young infants with growth failure and faltering.[2–4]

Malnutrition in children increases the risk of severe infections and triples the mortality risk from pneumonia, measles or diarrhoea.[5] Therefore, the current practice for children 6 months to 5 years of age with wasting is to prescribe routine antibiotics when they get into a nutrition programme, inpatient or outpatient.[5] However, current recommendations state that the same general medical care should be used for infants with severe acute malnutrition (SAM) who are less than 6 months of age as infants above 6 months of age, even though there is limited evidence to support this recommendation.[2] Furthermore, even though antibiotics are effective in children 6–59 months of age with SAM, this practice in infants has the potential to harm due to recently identified risks of antibiotic use in infancy, including the diminishment of infant gut microbiome,[6] future development of obesity, allergic disorders[7] and autoimmune disorders.[8] The urgency in more targeted management guidelines is further underscored by the physiological differences in renal and gastrointestinal function

in infants compared with older children.[8] The WHO recently started the guideline development process for preventing and treating wasting in children, including growth failure/faltering in infants under 6 months. This systematic review aimed to synthesise evidence from current literature on the effect of systematically given, routine use of antibiotics for infants less than 6 months of age with growth failure/faltering.

## OBJECTIVE
### Primary objective
In infants <6 months with growth failure/faltering, what are the effects of no routine antibiotics or different approaches (eg, types of antibiotics, doses) compared with routine antibiotics following treatment protocols in 2013 WHO guidelines[2] on the morbidity and mortality outcomes?

## METHODS AND ANALYSIS
This systematic review was conducted according to methods described in Cochrane Handbook[9] and reported using Preferred Reporting Items for Systematic Reviews and Meta-Analyses guidelines 2020.[10]

### Types of studies
We considered both individual and cluster randomised trials. We also considered non-randomised trials and cohort studies with a controlled arm. We excluded case–control studies, case reports, case series and commentaries.

### Population
The population of interest was infants under 6 months of age with growth failure/faltering. We considered the author's definitions because this age group has no standard definition of growth failure/faltering. We considered studies irrespective of whether they were done in community or hospital settings. We considered studies that included infants infected with HIV. We considered studies with low birth weight or preterm infants; however, we excluded studies on infants admitted to neonatal intensive care units. We excluded studies that only included infants with congenital anomalies.

### Intervention
We considered all antibiotic treatments given systemically, such as amoxicillin, Augmentin, cephalosporins and macrolides. We considered studies irrespective of dosage, frequency, duration or route of administration; however, topical application of antibiotics was not considered. We considered studies if antibiotics were given empirically at the time of diagnosis of growth failure or faltering, irrespective of the indication, for example, to treat an infection. We excluded studies where antibiotics were given to prevent wasting in otherwise healthy infants, specifically non-malnourished children or infants with no growth failure/faltering. We excluded studies where antibiotics

were given for other reasons, such as suspected serious bacterial infections in otherwise healthy infants, specifically non-malnourished children or infants with no growth failure/faltering.

### Comparison
The comparison group was routine antibiotics following treatment protocols detailed in the 2013 WHO guideline.[2]

### Outcomes
► Mortality (dichotomous outcome).
► Clinical deterioration (dichotomous outcome, defined by the development of any danger signs (obstructed breathing, respiratory distress, cyanosis, shock, severe anaemia, convulsion, severe dehydration, profuse watery diarrhoea, intractable vomiting and/or impaired consciousness)).
► Recovery from comorbidity (dichotomous outcome).
► Markers of intestinal inflammation-faecal calprotectin (continuous outcome).
► Markers of systemic inflammation-serum C reactive protein (continuous outcome).
► Hospital-acquired infections (dichotomous outcome).
► Non-response (eg, not achieving recovery within 4 months of initiating treatment) (dichotomous outcome).

All the primary analyses were considered at the longest follow-up. For the outcome of recovery from morbidity, the recovery could be recovery from diarrhoea, pneumonia, measles, etc.

## LITERATURE SEARCH
We conducted systematic electronic queries using key terms in multiple databases, including MEDLINE via PubMed, EMBASE, Web of Science, CINAHL, Scopus, LILACS, WHO Global Index Medicus and BIOSIS Previews. There were no search restrictions on outcomes, publication year, publication status or publication language. The search strategies for different databases are available in online supplemental appendix 1. The references of formerly published reviews and recently published studies were examined for potential inclusion. We also used The Cochrane Central Register for Controlled Trials and ISRCTN registry to identify studies currently underway. We also searched the websites of pertinent international agencies such as the WHO (including WHO's Reproductive Health Library, electronic Library of Evidence of Nutrition Actions and Global database on the Implementation of Nutrition Action), UNICEF, Global Alliance for Improved Nutrition, International Food Policy Research Institute, International Initiative for Impact Evaluation, Nutrition International, World Bank, USAID and affiliates (eg, FANTA, SPRING) and the World Food Programme. We searched the abstracts of major conferences, such as annual paediatric academic society meetings. Finally, we used the citation tracking

function of the included studies in PubMed to look for any other eligible studies.

## DATA EXTRACTION AND SYNTHESIS
### Selection of studies
Studies identified during the literature search were collected in an electronic reference manager EndNote[11] literature, and duplicated studies were removed. At least two authors screened study titles and abstracts to assess potential eligibility. Studies selected during this initial phase underwent a full-text review by two authors. The software Covidence[12] was used during the screening process. Disagreements were resolved by discussion, and the senior author on the team assisted as needed.

### Data extraction
We planned to extract the data for study region/country, study year, study type, intervention exposure (dose, duration, frequency), comparison, outcomes, population characteristics detailed in subgroup analysis and risk of bias. We also planned to extract information on the intervention's feasibility, acceptability, equity, and resource use and reported these data in separate tables. We planned to remove raw values for the number of events in the intervention and control group in case of dichotomous outcomes. To avoid reviewer bias, we decided a priori the order of preference for extracting outcomes when data were available in several formats.

### Studies with missing data
If a study was only available in an abstract, we contacted the authors for full text. If the full text could not be obtained from any sources, we considered the abstract if sufficient details of the study design and outcomes were available. We attempted to find the protocol of each potentially included study to assess the details of the methods. If the study protocol was not publicly available, we contacted the authors for the same. If the randomised trial results were published in more than one report, we considered all the publications related to that study as one study.

### Assessment of risk of bias in included studies
We aimed to evaluate the risk of bias from randomised controlled trials with the Cochrane risk of bias (ROB 2.0).[13] The risk of bias assessment according to ROB-2 is done for each outcome, not for a particular study.[13]

### Data synthesis
We planned to report the review findings both qualitatively and quantitatively. A narrative synthesis was considered to report all included studies' characteristics and results. A random-effects meta-analysis was planned when at least two studies possessed sufficient clinical and methodological uniformity for synthesis. The software RevMan was considered for statistical analysis.[14] We planned to assess the dichotomous outcomes using relative risk effect sizes and continuous outcomes with a mean difference and report with 95 % CIs.

### Assessment of heterogeneity
We aimed to analyse statistical heterogeneity in the pooled data using Tau,$^2$ $\chi^2$ and $I^2$ statistics. We also aimed to assess statistical heterogeneity through visual inspection of forest plots, using the $\chi^2$ test (assessing the p value) and calculating the Tau$^2$ and $I^2$ statistics. We considered it significant statistical heterogeneity when the p value was less than 0.1, the $I^2$ value exceeded 50%, and the inspection of forest plots showed substantial variability in the effect of the intervention. Finally, we considered subgroup analysis to identify reasons for eligible statistical heterogeneity.

### Assessment of reporting bias
We aimed to assess the publication bias of small studies using funnel plots and regression tests for funnel plot asymmetry when the meta-analysis included at least 10 studies.

### Subgroup analysis and investigation of heterogeneity
We planned the following subgroup analysis; however, none were possible due to a lack of studies.
► By different types/definitions of growth failure/faltering.
► Age at presentation (newborn (0–28 days), 1–3 months, 4–6 months).
► Gestational age: preterm birth (<37 weeks) versus full-term birth (>37 weeks).
► Birth weight: low birth weight (<2500 g) versus normal birth weight (>2500 g).
► HIV exposure: studies with participants exposed to HIV versus studies with no HIV exposure.
► Presentation: participants with oedema versus patricians with no oedema.
► Comorbidities: with or without comorbidities.
► Nutrition: babies breast feeding or non-breastfed babies.
► Location of the treatment: inpatient or outpatient/community.
► Dose of antibiotics.
► Duration of antibiotics: 7 days versus >7 days.
► Type of antibiotics.

### Sensitivity analysis
We planned to complete sensitivity analysis by the use of the model for meta-analysis.

### Rating of overall quality of evidence
The Grading of Recommendations Assessment, Development and Evaluation (GRADE) approach was considered to evaluate the overall certainty of evidence using the software GRADEpro.[15] The GRADE approach is a comprehensive framework used to assess the overall certainty of the evidence for an outcome using study characteristics such as study design, inconsistency, indirectness of evidence, risk of bias, publication bias and imprecision estimates.

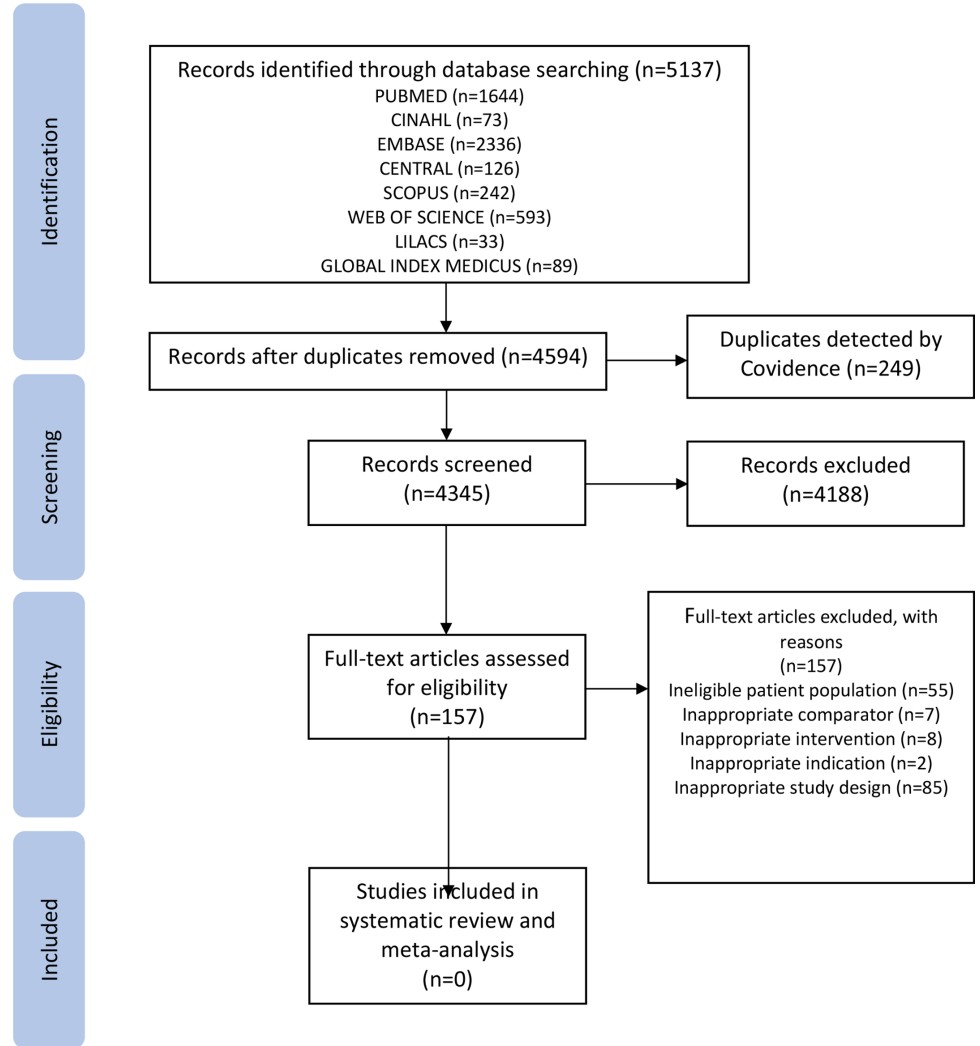

**Figure 1** Preferred Reporting Items for Systematic Reviews and Meta-Analyses flow diagram.

## Patient and public involvement

Patients and/or the public were not involved in this research's design, conduct, reporting or dissemination plans.

## RESULTS

### Literature search

We screened 5137 titles and abstracts; 157 eligible studies were screened for full-text reviews. None of the available studies qualified for inclusion in this review. Figure 1 shows the results of the literature search. The reasons for exclusions are available in the table of excluded studies (online supplemental document 1). In summary, 85 studies were excluded because of wrong study design, 55 were excluded due to ineligible patient population, 7 were excluded because of the ineligible comparison group, 8 were excluded because of ineligible intervention and 2 studies were excluded because of wrong indication.

## DISCUSSION

This systematic review aimed to assess the effect of routine use of antibiotics for the treatment of malnutrition in infants less than 6 months of age. Even though we reviewed more than 4000 titles and about 157 full-text studies, none of the studies qualified for inclusion in this review. The three key reasons to publish this work are to document our study question, to report the methodology transparently in enough detail so that it can be replicated in the future as needed, and to highlight the key gaps in research so that the research investigator can design studies for infants <6 months with malnutrition as the mortality risk is the highest due to malnutrition in this age group.

We followed the methodology of the Cochrane Collaboration to conduct this systematic review. A detailed protocol was prepared before the review process and was externally reviewed and registered publicly on PROSPERO's international database of prospective systematic reviews. The title and abstracts of the studies were screened independently by two study authors. We planned

to study the risk of bias using the revised Cochrane risk of bias 2.0 tool for each outcome within a study rather than determining the risk of bias based on all outcomes for a particular study. However, no eligible study was found to answer the clinical question in this review.

We noted two studies of great interest to our population of interest.[16][17] These studies were excluded because of the lack of a comparison group treated with antibiotics according to the current guideline of the WHO for treating wasting.[2] Both the studies included participants that fall in the age range (ie, <6 months) considered for this review; however, they also included participants beyond 6 months of age.[16][17] Both studies were randomised, double-blind, placebo-controlled trials. One study was conducted in Kenya[16] and another study was a multicounty trial.[17] The study from Kenya[16] used co-trimoxazole in the community settings to prevent mortality in severely malnourished children 2–59 months of age after being treated according to WHO protocol (both groups received antibiotics according to WHO protocol during the stabilisation period). Daily co-trimoxazole after initial treatment for severe malnutrition did not prevent mortality in children 2–59 months of age (HR 0.90, 95% CI 0.71 to 1.16). The results for infants 2 months to 5 months of age were similar to overall results.[16] The other study used biannual azithromycin in children 1–59 months of age irrespective of nutritional status[17] and showed a 13.5% reduction in mortality in the azithromycin group compared with placebo. A subgroup analysis for malnourished children showed similar results.[18] Two other studies could have been included; however, both included both nourished and malnourished participants, and we could not obtain the randomised data for our population of interest.[19][20] We also identified a recently completed trial[21] and requested the data for inclusion in this review; however, study investigators were submitting their results to a peer-review journal and are not yet willing to share those data.

### Implications for research

Future randomised studies, with adequate sample size of infants under 6 months with malnutrition and/or growth faltering, are needed to confirm the therapeutic effect of antibiotic treatment observed in children 6–59 months of age.[2] Additional data are also required to assess the appropriate antibiotic, dose, route, frequency and duration of antibiotic treatment. The safety profile regarding acute reactions, such as allergic reactions, gastrointestinal disturbance, etc, and long-term effects, such as effects on the gut microbiome and antibiotic resistance, also need further investigation.

### CONCLUSIONS

There is a paucity of evidence to assess the effect of routine use of antibiotics in infants less than 6 months of age with malnutrition. Future studies with large sample sizes are needed to evaluate the potential risks and benefits of antibiotics in malnourished children under 6 months of age.

**Acknowledgements** We are very thankful to Allison Daniel, Jaden Bendabenda and Zita Weise Prinzo, Kirrily De Polnay for their input in improving this review.

**Contributors** Conceptualisation: AI. Methodology: AI, FCC. Validation: AI, FCC, MF, MS. Formal analysis: AI, FCC. Resources: ET-S, AS, OT. Data curation: FCC, MF, MS. Writing: AI, FCC. Review and editing: AI, FCC, JKD, ZAB. All authors have read and agreed to the published version of the manuscript. AI is the guarantor of the manuscript.

**Funding** This work is funded by WHO grant no 202725572. WHO also provided technical support for this work.

**Disclaimer** We conducted literature searches, screening of titles, selection of studies, data extraction, and analysis according to the plan outlined in the protocol.

**Competing interests** None declared.

**Patient and public involvement** Patients and/or the public were not involved in the design, or conduct, or reporting, or dissemination plans of this research.

**Patient consent for publication** Not applicable.

**Ethics approval** Not applicable.

**Provenance and peer review** Not commissioned; externally peer reviewed.

**Data availability statement** All data relevant to the study are included in the article or uploaded as supplementary information. We provide the list of excluded studies and our search strategies. The results of the literature search are available on request.

**ORCID iDs**
Aamer Imdad http://orcid.org/0000-0002-7026-0006
Emily Tanner-Smith http://orcid.org/0000-0002-5313-0664
Jai K Das http://orcid.org/0000-0002-2966-7162

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
