## [Reviewer comments · BMJ Open]

ARTICLE DETAILS

TITLE (PROVISIONAL)	Routine antibiotics for infants less than six months of age with growth failure/faltering. A systematic review
AUTHORS	Imdad, Aamer; Chen, Fanny F; François, Melissa; Sana, Momal; Tanner-Smith, Emily; Smith, Abigail; Tsistinas, Olivia; Das, Jai K.; Bhutta, Zulfiqar

VERSION 1 – REVIEW

REVIEWER	Hamilton, Seona NHS Greater Glasgow and Clyde, Library Network
REVIEW RETURNED	01-Feb-2023

GENERAL COMMENTS	My review of this manuscript is limited to the search process and adherence to systematic review processes and reporting guidelines. I am not qualified to comment on the validity of the findings. I would recommend a further review of the final text to check and amend grammatical errors - these are very minor and shouldn't take long to correct, but do detract from the readability of the final article. The search process was comprehensive with an appropriate range of bibliographic database sources plus use of supplementary search methods to ensure that studies not found as part of the database search could be located and included. The search strategies as reported are well conducted: - search question broken down into appropriate search topics (infants/children; antibiotics) and the search clearly structured- all relevant database-specific controlled vocabulary terms included- appropriate use of free text terms + synonyms- appropriate use of proximity & truncation symbols- terms correctly combined using AND/OR- search correctly translated between search interface providers My only query re: choice of search terms is the non-inclusion of individual drug names (e.g. amoxicillin, cephalosporin) in the lists of free text search terms. As the subject heading searches for antibiotics were exploded this may have mirrored the searches more closely, particularly for databases where subject indexing wasn't available. Again, it's unlikely that this will have changed the outcome of your review given your use of supplementary search methods, but it might be worth testing what difference it would have made, or if a deliberate decision was made not to include
--

	individual drug names to detail the rationale in the methods section. I would recommend not attempting to exclude Medline records when conducting searches in other databases. Sometimes differences in indexing can mean that you find relevant results in, for e.g. Web of Science, that were missed in Medline. As your database results were going through a deduplication process anyway, there was very little effort saved in removing these results at the search stage compared to the (albeit very small) risk of missing relevant papers. However, given the number of sources searched and the use of supplementary search methods I think it is unlikely to have affected the results. Your search strategies in the draft pdf are very difficult to read, and impossible to run without correcting first, due to html character codes appearing instead of some punctuation. I would recommend correcting this before publication. The reporting of the search and screening process conforms to the PRISMA guidelines and the PRISMA 2020 checklist is included. I'm happy that the methods used demonstrate that the authors have taken all reasonable efforts to identify research to answer that question at the point of the search.
--	--

REVIEWER	Turi, Kedir Vanderbilt University, Medicine
REVIEW RETURNED	03-Feb-2023

GENERAL COMMENTS	Major:  - Authors need to say What value would this systematic review add to the literature other than saying no relevant study to review? - Can the authors put more details about the studies in table 1? Minor: Check uses of parenthesis and punctuations especially in outcome definition section
--

REVIEWER	Theodosiou, Anastasia University of Southampton
REVIEW RETURNED	20-Feb-2023

GENERAL COMMENTS	This is an interesting, worthwhile and timely topic for review, and the clear and concise introduction sets out the rationale for review very nicely. The authors did not identify any studies for inclusion, although the inclusion and exclusion criteria were not inappropriately stringent. Indeed, the authors rightly draw attention to the lack of available evidence in infant aged less than 6 months. However, the lack of any studies for inclusion does limit the suitability for publication in the review's present form. Given the hard work the authors have already done in reviewing the literature, it would be useful to add a section describing the evidence that is available, even though it did not meet the inclusion criteria defined. The authors begin to do this in the discussion, but the reader is left wondering whether any lessons could have been learnt from the other excluded studies (including observational, case-controls, and trials without comparators). Provided it is made clear to the reader that these studies did not meet the inclusion criteria defined at the outset, I think it would add
---

	value by showing what the extent of available (albeit limited) evidence is at present. It is also unclear from the discussion whether the authors feel that there is insufficient evidence to justify the application of the WHO guidance regarding antibiotic use in malnutrition to infants under 6 months, or whether this guidance should still be followed despite the paucity of evidence. Also, there is a great deal of methodology included pertaining to what would have been done if studies were identified for inclusion – I wonder whether the “would have done” methods should be grouped together as a supplement, rather than included in the main text, given that these methods weren’t actually used? Overall, this is a well-written and interesting paper, but I think it needs more in way of results and content before publication can be recommended.
--	--

VERSION 1 – AUTHOR RESPONSE

Reviewer: 1

Ms. Seona Hamilton, NHS Greater Glasgow and Clyde

Comments to the Author:

My review of this manuscript is limited to the search process and adherence to systematic review processes and reporting guidelines. I am not qualified to comment on the validity of the findings.

Author’s response: Thank you, we appreciate your time to review our work

I would recommend a further review of the final text to check and amend grammatical errors - these are very minor and shouldn’t take long to correct, but do detract from the readability of the final article.

Author’s response: Thank you for pointing this out. We have revised our manuscript and fixed any grammatical errors.

The search process was comprehensive with an appropriate range of bibliographic database sources plus use of supplementary search methods to ensure that studies not found as part of the database search could be located and included.

The search strategies as reported are well conducted:

- search question broken down into appropriate search topics (infants/children; antibiotics) and the search clearly structured
- all relevant database-specific controlled vocabulary terms included
- appropriate use of free text terms + synonyms
- appropriate use of proximity & truncation symbols
- terms correctly combined using AND/OR
- search correctly translated between search interface providers

Author’s response: Thank you, we appreciate your time to look at our work in such detail and happy to know that we covered all the necessary steps in literature search.

My only query re: choice of search terms is the non-inclusion of individual drug names (e.g. amoxicillin, cephalosporin) in the lists of free text search terms. As the subject heading searches for antibiotics were exploded this may have mirrored the searches more closely, particularly for databases where subject indexing wasn’t available. Again, it’s unlikely that this will have changed the outcome of your review given your use of supplementary search methods, but it might be worth

testing what difference it would have made, or if a deliberate decision was made not to include individual drug names to detail the rationale in the methods section.

Author's response: Thank you for this important comment. It was intentional to not include the name of specific antibiotics as that would limit the sensitivity of the search. Also, there is no standard of care for infants less than six months (the evidence comes from children 6-59 months and is applied to infants less than six months of age), so we want to capture any antibiotics studied in this age group.

I would recommend not attempting to exclude Medline records when conducting searches in other databases. Sometimes differences in indexing can mean that you find relevant results in, for e.g. Web of Science, that were missed in Medline. As your database results were going through a deduplication process anyway, there was very little effort saved in removing these results at the search stage compared to the (albeit very small) risk of missing relevant papers. However, given the number of sources searched and the use of supplementary search methods I think it is unlikely to have affected the results.

Author's response: Thank you for pointing this out. We agree that Medline records could have been included while looking at other databases, however we searched multiple sources and feel confident that no relevant study was missed. We also had additional support from experts at WHO who also reported that they are not aware of any other studies beyond our work that could have been included in the review.

Your search strategies in the draft pdf are very difficult to read, and impossible to run without correcting first, due to html character codes appearing instead of some punctuation. I would recommend correcting this before publication.

Author's response: Thank you, we agree and revised the size of the font of the search strategy in appendix 1 to improve the readability. The appearance of punctuation sometimes happens when the text is being copied from the PDF.

The reporting of the search and screening process conforms to the PRISMA guidelines and the PRISMA 2020 checklist is included.

Author's response: Thank you

I'm happy that the methods used demonstrate that the authors have taken all reasonable efforts to identify research to answer that question at the point of the search.

Author's response: Thank you, we are pleased to know that our search methods were appropriate

Reviewer: 2

Dr. Kedir Turi, Vanderbilt University

Comments to the Author:

Major:

- Authors need to say What value would this systematic review add to the literature other than saying no relevant study to review?

Author's response: Thank you and we appreciate the time from the respected reviewer to assess our work. This work was part of an exercise to synthesize the evidence on treatment of severe malnutrition so that WHO can produce the guidelines for treatment of malnutrition in developing countries. The key reasons to publish this work is to document the clear and comprehensive

methodology for this work so that it can be replicated in the future as needed, and to highlight the key gaps in research so that potential research investigator can design the studies for infants < 6 months as the mortality risk is the highest due to malnutrition in this age group. We have now added the following text to discussion to elaborate this further

“The three key reasons to publish this work is to document our study question; to report the methodology in a transparent manner in enough details so that it can be replicated in the future as needed, and to highlight the key gaps in research so that research investigator can design studies for infants < 6 months with malnutrition as the mortality risk is the highest due to malnutrition in this age group.”

- Can the authors put more details about the studies in table 1?

Author's response: Thank you, This is a very important comment. We agree and provided descriptions on the studies that were almost included in our review but excluded due to lack of appropriate comparison or study population in our discussion section as follows

“We noted two studies that were of great interest to our population of interest^{16,17}. These studies were excluded because of lack of a comparison group that was treated with antibiotics according to current guideline of World health Organization for treatment of wasting.² Both the studies included participants that fall in the age range (i.e. < 6 months) considered for this review however they also included participants beyond six months of age^{16,17}. Both studies were randomized double-blind placebo-controlled trials. One study was conducted in Kenya¹⁶ and another study was a multicounty trial¹⁷. The study from Kenya ¹⁶ used co-trimoxazole in the community settings for prevention of mortality in severely malnourished children 2-59 months of age after they were treated according to WHO protocol (both the groups received antibiotics according to WHO protocol during the stabilization period). Daily Co-trimoxazole after initial treatment for severe malnutrition did not lead to prevention of mortality in children 2-59 months of age (Hazard ratio 0.90, 95% CI 0.71-1.16). The results for infants 2 months to 5 months of age were similar to overall results¹⁶. The other study used biannual azithromycin in children 1-59 months of age irrespective of nutritional status¹⁷ and showed a 13.5 % reduction in mortality in the azithromycin group compared to placebo. A subgroup analysis for malnourished children showed similar results¹⁸. There were two other studies that could have been included; however, both of these studies included both nourished and malnourished participants and we could not obtain the randomized data for our population of interest^{19,20}. We also identified a recently completed trial²¹ and requested the data for inclusion to this review, however study investigators were in process of submission of their results to a peer review journal and are not yet willing to share those data.”

As we discussed these studies in the discussion section, we avoided additional text in the result section.

Minor: Check uses of parenthesis and punctuations especially in outcome definition section

Author's response: Thank you, we agree and reviewed this section and made the edits as suggested.

Reviewer: 3

Dr. Anastasia Theodosiou, University of Southampton

Comments to the Author:

This is an interesting, worthwhile and timely topic for review, and the clear and concise introduction sets out the rationale for review very nicely. The authors did not identify any studies for inclusion, although the inclusion and exclusion criteria were not inappropriately stringent. Indeed, the authors rightly draw attention to the lack of available evidence in infant aged less than 6 months. However, the lack of any studies for inclusion does limit the suitability for publication in the review's present form.

Author's response: Thank you, we highly appreciate the time from the respected reviewer to assess our work. We addressed the comment related to the importance of this work in response to reviewer 2 and we think the publication of this work in a peer review journal will add credibility to guideline development work by the World Health Organization who commissioned this work for the same purpose.

Given the hard work the authors have already done in reviewing the literature, it would be useful to add a section describing the evidence that is available, even though it did not meet the inclusion criteria defined. The authors begin to do this in the discussion, but the reader is left wondering whether any lessons could have been learnt from the other excluded studies (including observational, case-controls, and trials without comparators). Provided it is made clear to the reader that these studies did not meet the inclusion criteria defined at the outset, I think it would add value by showing what the extent of available (albeit limited) evidence is at present.

Author's response: Thank you, we agree and described some of the studies with control groups that could have been included but were excluded due to inappropriate comparison or lack of appropriate population. For the rest of the study designs, we were directed by WHO to not consider the evidence from studies without a control as it is extremely difficult to make an association of effectiveness of the intervention in these studies and this type of evidence is considered of very 'low quality' in the GRADE methodology. We therefore did not synthesize the data from about 150 studies for the same reasons.

It is also unclear from the discussion whether the authors feel that there is insufficient evidence to justify the application of the WHO guidance regarding antibiotic use in malnutrition to infants under 6 months, or whether this guidance should still be followed despite the paucity of evidence.

Author's response: Thank you for this very important comment. As a systematic reviewer, we avoided making a comment/recommendation in our discussion in favor or against the use of antibiotics in malnourished infants less than six months of age. This is because the guidelines groups considered multiple other aspects of intervention beyond the efficacy and safety of the intervention such as cost, equity, feasibility, patients values and preferences. We therefore left it to WHO to make recommendations in this regard where there will be content experts, methodologists, patient representatives and stakeholders as part of the guideline panel who will make the decision with consensus.

Also, there is a great deal of methodology included pertaining to what would have been done if studies were identified for inclusion – I wonder whether the “would have done” methods should be grouped together as a supplement, rather than included in the main text, given that these methods weren't actually used? Overall, this is a well-written and interesting paper, but I think it needs more in way of results and content before publication can be recommended.

Author's response: Thank you for these important observations. We agree that some of the text might be reduced, however, it is intentional to keep the detailed methods so that this work can be replicated as needed in the future.